# Circulation of *Babesia* Species and Their Exposure to Humans through *Ixodes ricinus*

**DOI:** 10.3390/pathogens10040386

**Published:** 2021-03-24

**Authors:** Tal Azagi, Ryanne I. Jaarsma, Arieke Docters van Leeuwen, Manoj Fonville, Miriam Maas, Frits F. J. Franssen, Marja Kik, Jolianne M. Rijks, Margriet G. Montizaan, Margit Groenevelt, Mark Hoyer, Helen J. Esser, Aleksandra I. Krawczyk, David Modrý, Hein Sprong, Samiye Demir

**Affiliations:** 1Centre for Infectious Disease Control, National Institute for Public Health and the Environment, 3720 BA Bilthoven, The Netherlands; ryanne.jaarsma@rivm.nl (R.I.J.); arieke.docters.van.leeuwen@rivm.nl (A.D.v.L.); manoj.fonville@rivm.nl (M.F.); miriam.maas@rivm.nl (M.M.); frits.franssen@rivm.nl (F.F.J.F.); aleksandra.i.krawczyk@gmail.com (A.I.K.); hein.sprong@rivm.nl (H.S.); demirsamiye@gmail.com (S.D.); 2Dutch Wildlife Health Centre, Utrecht University, 3584 CL Utrecht, The Netherlands; m.kik@uu.nl (M.K.); j.m.rijks@uu.nl (J.M.R.); m.montizaan@uu.nl (M.G.M.); 3Diergeneeskundig Centrum Zuid-Oost Drenthe, 7741 EE Coevorden, The Netherlands; margitgroenevelt@zod.nl; 4Veterinair en Immobilisatie Adviesbureau, 1697 KW Schellinkhout, The Netherlands; m.hoyer-via@planet.nl; 5Wildlife Ecology & Conservation Group, Wageningen University, 6708 PB Wageningen, The Netherlands; helen.esser@wur.nl; 6Laboratory of Entomology, Wageningen University, 6708 PB Wageningen, The Netherlands; 7Institute of Parasitology, Biology Centre CAS, 370 05 Ceske Budejovice, Czech Republic; modrydav@gmail.com; 8Department of Botany and Zoology, Faculty of Science, Masaryk University, 611 37 Brno, Czech Republic; 9Department of Veterinary Sciences/CINeZ, Faculty of Agrobiology, Food and Natural Resources, Czech University of Life Sciences Prague, 165 00 Prague, Czech Republic

**Keywords:** babesiosis, *Ixodes ricinus*, sylvatic cycle, zoonoses, disease risk, One Health

## Abstract

Human babesiosis in Europe has been attributed to infection with *Babesia divergens* and, to a lesser extent, with *Babesia venatorum* and *Babesia microti,* which are all transmitted to humans through a bite of *Ixodes ricinus*. These *Babesia* species circulate in the Netherlands, but autochthonous human babesiosis cases have not been reported so far. To gain more insight into the natural sources of these *Babesia* species, their presence in reservoir hosts and in *I. ricinus* was examined. Moreover, part of the ticks were tested for co-infections with other tick borne pathogens. In a cross-sectional study, qPCR-detection was used to determine the presence of *Babesia* species in 4611 tissue samples from 27 mammalian species and 13 bird species. Reverse line blotting (RLB) and qPCR detection of *Babesia* species were used to test 25,849 questing *I. ricinus*. Fragments of the 18S rDNA and cytochrome c oxidase subunit I (COI) gene from PCR-positive isolates were sequenced for confirmation and species identification and species-specific PCR reactions were performed on samples with suspected mixed infections. *Babesia microti* was found in two widespread rodent species: *Myodes glareolus* and *Apodemus sylvaticus*, whereas *B. divergens* was detected in the geographically restricted *Cervus elaphus* and *Bison bonasus*, and occasionally in free-ranging *Ovis aries*. *B. venatorum* was detected in the ubiquitous *Capreolus capreolus*, and occasionally in free-ranging *O. aries*. *S*pecies-specific PCR revealed co-infections in *C. capreolus* and *C. elaphus,* resulting in higher prevalence of *B. venatorum* and *B. divergens* than disclosed by qPCR detection, followed by 18S rDNA and COI sequencing. The non-zoonotic *Babesia* species found were *Babesia capreoli*, *Babesia vulpes, Babesia* sp. deer clade*,* and badger-associated *Babesia* species. The infection rate of zoonotic *Babesia* species in questing *I. ricinus* ticks was higher for *Babesia* clade I (2.6%) than *Babesia* clade X (1.9%). Co-infection of *B. microti* with *Borrelia burgdorferi* sensu lato and *Neoehrlichia mikurensis* in questing nymphs occurred more than expected, which reflects their mutual reservoir hosts, and suggests the possibility of co-transmission of these three pathogens to humans during a tick bite. The ubiquitous spread and abundance of *B. microti* and *B. venatorum* in their reservoir hosts and questing ticks imply some level of human exposure through tick bites. The restricted distribution of the wild reservoir hosts for *B. divergens* and its low infection rate in ticks might contribute to the absence of reported autochthonous cases of human babesiosis in the Netherlands.

## 1. Introduction

Babesiosis in humans and domesticated animals is caused by infection with tick-borne apicomplexan parasites of the genus *Babesia* [1]. These parasites are also called piroplasms due to their pear-shaped appearance when replicating in erythrocytes. Initially, three lineages of piroplasms were defined, based on character-state definitions, but more recent phylogenic analyses showed that Piroplasmida comprises of approximately six lineages, of which *Babesia* species represent at least three distinct clades [2,3,4]. The clade terminology throughout this paper follows the most recent study [3]. The *B. microti-*like lineage, clade I, includes *Babesia microti, Babesia vulpes, Babesia felis,* and *Babesia* species recently identified in badgers [5,6]. *Babesia* species from clade X, originally designated as *Babesia* sensu stricto, includes *Babesia divergens, Babesia venatorum* (formerly called EU1–3), *Babesia capreoli, Babesia canis,* and *Babesia* sp. deer clade, sometimes referred to as the European variant of *Babesia odocoilei* [7]. Particularly *Babesia* species from clade X are of economic importance in the livestock industry [8,9]. In addition, both clades comprise some species that are considered zoonotic and can affect human health [10,11]. However, it should be noted that within the designated *Babesia* species, genetic variants exist that might confer their zoonotic potential [12,13,14,15]. In the Netherlands, *B. microti, B. divergens, B. venatorum, B. capreoli, Babesia motasi, Babesia caballi,* and *B. canis* have been found in wildlife, domestic animals or zoo animals [16,17,18,19,20,21]. Whether zoonotic variants are present, and what other *Babesia* species circulate in the Netherlands is not known.

The majority of human cases of babesiosis have been caused by *B. microti* in the United States, with an (under)estimated incidence of 2000 cases per year [22], followed by Canada and China [11]. In contrast, the number of reported cases in Europe has remained low [23]. To date, approximately fifty cases have been reported in Europe [24,25,26,27]. The majority of these cases have been attributed to *B. divergens*, three to *B. venatorum* and two cases to *B. microti* [28,29]. Persistent infections with *B. divergens* are mostly confined to asplenic patients, and are characterized by sepsis, severe anemia, hemoglobinuria, and jaundice due to hemolysis. Persistent infections with the other two *Babesia* species appear to be less severe, although the reported cases were also of asplenic or immunocompromised patients [13,30]. Despite the low clinical incidence, significant seroprevalence rates in humans have been recorded in many parts of Europe, indicating that exposure or (endured) infection with *Babesia* species is not a rare event [31,32,33,34,35,36]. A first possible explanation for these findings is that the European *Babesia* species do not give rise to persistent infections in immunocompetent humans, although they may cause seroconversion [37]. A second possible explanation may be that human babesiosis is underdiagnosed due to a milder disease course and non-characteristic symptoms in immunocompetent patients, lack of awareness about the disease, and the difficulty in making a diagnosis [27,38]. Although molecular evidence of infection with *B. divergens* in humans has been found in the Netherlands, no clinical cases of autochthonous babesiosis have been reported thus far [39].

Many hard ticks (Ixodidae) have been identified as vectors of *Babesia* species in the literature, but most of these are based on a perceived association with the disease rather than on an objective demonstration of transmission [40,41]. In Europe, *Ixodes ricinus* ticks serve as the principal vectors of various emerging human tick-borne pathogens, and are also considered to transmit *B. microti, B. divergens* and *B. venatorum* to humans [24,42]. *Ixodes ricinus* feed only once per life stage, and *Babesia* species have developed the ability to persist through successive tick developmental stages, referred to as transstadial transmission. An additional perpetuation strategy of *Babesia* species from clade X is transovarial transmission, allowing for the spread of the parasite from a single maternal tick to thousands of offspring [43]. Though several studies already investigated the presence and distribution of *Babesia* species in *I. ricinus* in the Netherlands, the accuracy of these studies has been limited due to the relatively low number of ticks tested in combination with the low infection rates of *Babesia* species [16,44]. A third explanation for the lack of reported cases of babesiosis in the Netherlands might therefore be the low infection rate of zoonotic *Babesia* species in *I. ricinus,* hence a low human exposure to *Babesia* species through tick bites.

The infection rate of *Babesia* species in *I. ricinus* ticks depends on the abundance and spread of competent vertebrate hosts [45,46,47]. A competent reservoir host is a vertebrate from which *I. ricinus* can become infected with *Babesia* species and vice versa [48]. The identification of competent reservoir hosts for zoonotic *Babesia* species is key to determine the disease hazard [49,50]. Well-documented hosts for zoonotic *Babesia* species in Europe are cattle (*B. divergens*), roe deer (*B. venatorum*), and small mammals (*B. microti*). Although some deer species have been incriminated by PCR as hosts of *B. divergens* in several studies, their role in the transmission of zoonotic *B. divergens* remains poorly understood [12,13]. To further complicate the matter, *Babesia* species might have partly overlapping competent hosts [51,52,53], and other tick species, such as *Ixodes trianguliceps*, might be involved in sylvatic cycles as well [24,54,55,56]. Moreover, the abundance of non-competent hosts may reduce the pathogen prevalence in ticks via a dilution effect [57,58]. A fourth explanation for the lack of reported cases of babesiosis in the Netherlands might be a limited distribution of competent reservoir hosts, which would result in a low or only localized exposure to zoonotic *Babesia* species [59].

In this cross-sectional study, we tested for the presence of different *Babesia* species in different vertebrates to determine their role as potential reservoir hosts in the Netherlands. Furthermore, we evaluated the relative abundance of *Babesia* species in questing ticks and analyzed the probability of co-infections with other tick-borne pathogens (TBPs) as a basis to understand the hazard and exposure of each species known to cause disease.

## 2. Results

### 2.1. Babesia in Wildlife

#### 2.1.1. Babesia Sensu Stricto (Clade X)

A total of 4611 vertebrate samples were tested by qPCR for the presence of *Babesia* DNA. Using qPCR for the specific detection of the *Babesia* sensu stricto*-*clade (clade X), infection was found in the cloven-hooved mammals *Bison bonasus* (21%; n = 19), *Capreolus capreolus* (85%; n = 608), *Cervus elaphus* (64%; n = 147), *Dama dama* (9%; n = 100), *Ovis aries* (2%; n = 634), but not in *Bos taurus* (n = 116), *Capra aegagrus* (n = 5) and *Sus scrofa* (n = 111) (Table 1). *Babesia* sensu stricto*-*clade was not detected in any of the birds (n = 99), carnivores (n = 812), lagomorphs (n = 238), rodents (n = 1550), horses (n = 15), hedgehogs (32) or moles (n = 125) tested (Table 1). The qPCR-positive samples were typed to the species level based on a fragment of the 18 S rRNA gene obtained from conventional PCR (Table 1). The 18 S rRNA fragment of *B. capreoli* (FJ944828) is only two nucleotides different from *B. divergens* (AY046576), namely at positions 631 and 663 [51]. Typing was successful only for part of the samples: out of the 635 qPCR-positive *Babesia* sensu stricto samples, 366 DNA sequences were retrieved. *Babesia capreoli* was identified in 291 *C. capreolus* and in four *D. dama* samples, *B. venatorum* in three *C. capreolus* and three *O. aries,* and *B. divergens* was successfully typed from 26 *C. elaphus*, two *B. bonasus,* and two *O. aries* tissue samples (Table 1).

Remarkably, 65 of the 18 S rRNA sequences from *C. capreolus* (roe deer) contained double peaks in the sequence trace files, resulting in ambiguous nucleotides (not shown). Fifteen samples were probably mixed infections with *B. capreoli* and *B. venatorum* and in eight sequences *B. capreoli* and *B. divergens* could not be differentiated because of the presence of one or two ambiguous nucleotides at positions 631 or 663 (Table 1, designated as 23 ambiguous 18 S rRNA sequences). The 518 qPCR-positive *C. capreolus* samples were therefore also typed to the species level using DNA sequences obtained from PCR on a fragment of the cytochrome c oxidase subunit I (COI) gene. The genetic diversity of the COI fragment is higher than that of the 18 S rRNA fragment, and can therefore unequivocally distinguish between the different *Babesia* species. However, the success rate of COI typing is lower than that of 18S rRNA. With 18 S rRNA typing, 316 samples yielded a *Babesia* sequence compared to 158 samples with cytochrome c oxidase subunit I (COI) sequences. From these COI sequences, 134 were designated as *B. capreoli,* eight as *B. venatorum,* and 16 sequences could not be assigned because of the large number of ambiguous nucleotides (not shown).

*Capreolus capreolus* derived sequences that presented double peaks in some of the sequence trace files were suspected of mixed infections. To be able to estimate their occurrence and to identify the *Babesia* species involved, these samples were subjected to four separate PCRs with species-specific primers. Each PCR was designed for the specific detection of *B. venatorum*, *B. divergens*, *B. capreoli* or *Babesia* sp. deer clade. A mixed infection in a *C. capreolus* sample was identified when two or three of the specific PCRs yielded a product of the right size. From all the *C. capreolus* samples, 290 were randomly selected and tested. Infection with *B. capreoli* was found in 89% of the samples (n = 258), *B. venatorum* was detected in 46% of the samples (n = 132) and *B. divergens* was not detected in any of the tested samples. Infections with both *B. capreoli* and *B. venatorum* were found in 44% of the 290 samples (n = 128).

As mixed infections were also suspected in *C.*
*elaphus* (red deer), 84 randomly selected samples were subjected to the species-specific PCRs. Out of 77 positive samples, *B. divergens* was detected in 80% of the samples (n = 67), *Babesia sp. deer clade* was detected in 74% of the samples (n = 62), *B. venatorum* was not detected in any of the tested samples. Infection with both *B. divergens* and *Babesia sp. deer clade* was found in 71% of the samples (n = 60).

#### 2.1.2. Babesia Clade I

Using qPCR detection of DNA from the *B. microti-like-*clade, infection was found in *Meles meles* (84%; n = 128), *Nyctereutes procyonoides* (29%; n = 7), *Vulpes vulpes* (72%; n = 173)*,*
*Apodemus sylvaticus* (0.8%; n = 634), *Microtus arvalis* (3%, n = 100) and in *Microtus glareolus* (6%; n = 405) (Table 1). DNA from the *B. microti-like-*clade was not detected in any of the ungulates (n = 1740), birds (n = 99), lagomorphs (n = 238), horses (n = 15) hedgehogs (n = 32), or moles (n = 125) tested, neither in several other mustelid or rodent species (Table 1). qPCR-positive samples were typed to the species level using DNA sequences obtained from a conventional PCR fragment of the 18 S rRNA gene (Table 1), a fraction of which were confirmed by sequencing of the COI gene (Appendix A). Based on the 18 S rRNA sequences, which were successfully retrieved for part of the samples, *B. microti* was identified in *M. glareolus* (n = 12), *A. sylvaticus* (n = 2), and *B. vulpes* was found in *V. vulpes* (n = 58) and *N. procyonoides* (n = 2), whereas three different *Babesia* badger*-*type sequences were identified in *M. meles* (n = 53).

### 2.2. Geographic Distribution of Wildlife

The reservoir hosts for *B. divergens*, *C. elaphus* (140 blocks) and *B. bonasus* (9 blocks)*,* were restricted to less than 10% of the surface area of the country (Figure 1A,B). *Capreolus capreolus* (1483 blocks) is the main reservoir host for *B. capreoli* and *B. venatorum,* and has been observed in more than 90% of the surface area of the country (Figure 1C). The distribution of *D. dama* (279 blocks)*,* which carried *B. capreoli* and *Babesia* sp. deer clade-EU is also dispersed, but only in ±20% of the country (Figure 1D). The reservoir hosts for *B. microti,* namely *A. sylvaticus* (1549 blocks) and *M. glareolus* (1291 blocks) (Figure 2A,B) are widely spread and their distribution covers almost the whole of the Netherlands. *V. vulpes* (1509 blocks) and *N. procyonoides* (145 blocks) (Figure 2C,D), the reservoir hosts for *B. vulpes*, have been observed in more than 90% of the surface area of the country. Lastly, *M. meles* (760 blocks) (Figure 2E), which are reservoir hosts for the *Babesia* badger-types, were observed in ~50% of the country, mostly in the eastern half of the country. It appears to be absent in areas with sea clay soil and dunes (Figure 1). *Babesia* clade I was also found in the widely spread *M*. *arvalis* (Figure 2F); however, these sequences could not be typed to the species level.

### 2.3. Babesia Species in Ixodes ricinus

Questing ticks from various field studies in the Netherlands and one Belgian study were screened for *Babesia* clade I and X by reverse line blot hybridization assay (RLB) until 2012, and thenceforth by qPCR (Table 2 and Table 3).

*Babesia* sensu stricto*-*clade DNA was detected in 1.9% (n = 25,849) of the questing *I. ricinus* nymphs collected in the Netherlands (Table 2). All qPCR-positive samples (n = 489) were typed to the species level by using PCR and sequencing a fragment of the 18 S rRNA gene. Based on the retrieved 18 S rRNA sequences (n = 226), *B. venatorum* was identified in 0.8% (n = 210), *B. capreoli* in 0.04% (n = 11), *B. divergens* in 0.01% (n = 4), and the European variant of *Babesia* sp. deer clade in <0.01% (n = 1) of the ticks (Table 2). The four *B. divergens-*positive ticks were from areas where either *B. bonasus* or *C. elaphus* were present (not shown). *Babesia microti-like-*clade DNA was detected in 2.6% (n = 18,626) of the questing *I. ricinus* nymphs collected in the Netherlands (Table 3). Typing by 18 S rRNA sequencing was successful for a fraction of the qPCR-positive samples (n = 45) identifying all of them as *B. microti.*

In addition to detection of *Babesia*, 8831 nymphs collected from the Nijverdal and Rodent studies (Table 2 and Table 3) were analyzed for the presence of *Borrelia burgdorferi* sensu lato, *Neoehrlichia mikurensis, Borrelia miyamotoi,* and *Anaplasma phagocytophilum.* Co-infection of *Babesia* clade X with *A. phagocytophilum* occurred significantly less than expected (Table 4). Co-infection of *Babesia* clade I with *B. burgdorferi* sl and *N. mikurensis* occurred significantly more than expected randomly (Table 4).

## 3. Discussion

We have described the presence of different *Babesia* species in over 4000 samples from various vertebrate hosts in order to determine their role as potential reservoirs for zoonotic *Babesia* species in the Netherlands. Moreover, we have shown that some ungulate species may host more than one *Babesia* species concomitantly.

*Babesia* sensu stricto species *B. capreoli, B. venatorum, B. divergens,* and *Babesia* sp. deer clade were detected in cloven-hooved mammals as has been reported in other European countries [64,65,66,67,68,69,70]. Of the 635 *Babesia* sensu stricto positive samples, 366 were successfully typed to the species level. *B. venatorum* was found in a fraction of clade X positive *C. capreolus* samples (5/518) and of *O. aries* positive samples (3/10). *B. venatorum* has been detected in European mammals [64,65,66,67], mainly in *C. capreolus*, with a prevalence ranging from 0.4% in Italy [66] and 1.6% in the Czech Republic [7] to 26.0% in Germany [65]. *Babesia divergens* was found in *C. elaphus* (26/94) and *B. bonasus* (2/4), and to a lesser extent in *O. aries* (2/10). Other studies have reported a prevalence in *C. elaphus* ranging from 2.6% in Switzerland [53] to 33% in Poland [71], where the prevalence in *B. bonasus* has also been found to be above 30% [72]. However, the numbers in the current study are an underestimation, as typing to the species level using Sanger sequencing was only successful for a fraction of qPCR positive samples. This can be corroborated by the results of the species-specific PCR reactions that were performed for a selection of the samples. Co-infections in *C. capreolus* and *C. elaphus* samples revealed *B. venatorum* and *B. divergens* sequences were masked behind *B. capreoli* and *Babesia* sp. deer clade sequences, respectively. Thus, a higher prevalence of the zoonotic *Babesia* was uncovered (*B. venatorum* in 46% of *C. capreolus* samples and *B. divergens* in 80% of *C. elaphus* samples). Evidence showing one host can harbor more than one *Babesia* species, has been reported previously, in Czech, Norwegian, Swiss, and Austrian deer [7,53,73,74]. These results highlight the limitation of relying upon a single qPCR with successive PCR and Sanger sequencing, as this approach cannot detect concomitant microorganisms of the same genus as accurately as species-specific PCR does. Moreover, these results should encourage the re-evaluation of past surveys and the approach of future ones when dealing with TBPs of various zoonotic species.

Despite the high prevalence of *B. divergens* in *C. elaphus* in the Netherlands, the prevalence in questing nymphs was low (<0.01%) in accordance with previous studies from the Netherlands and other European countries [16,75,76,77,78,79,80,81]. Although *B. divergens* has been detected in *I. ricinus* nymphs and larvae before and has been experimentally acquired and transmitted by them [82,83], other studies have detected *B. divergens* predominantly in adult ticks [84]. Moreover, in vivo studies showed only adult ticks can successfully acquire the piroplasm from infected cattle [85] and observations suggesting large mammals are mostly parasitized by adult ticks [37] show these may play a greater role in the acquisition of *Babesia* clade X. As *B. divergens* is transmitted both transovarially and transstadially [85,86], piroplasm loads may be higher in larvae than nymphs. These variations could be significant and confound molecular detection methods of a certain sensitivity when loads are low, and it has been suggested that testing salivary glands and not whole ticks may lead to more accurate results [83]. Moreover, *Babesia* reproduction is induced by feeding [37] and most studies screen questing ticks. Lastly, the low prevalence in ticks may reflect that neither acquisition by adults, nor transovarial and transstadial transmission have been found to be 100% efficient [83,85].

Surprisingly, an analysis of co-infections in a subset of the questing ticks revealed a negative correlation between Clade X and *A. phagocytophilum*, the etiologic agent of human granulocytic anaplasmosis (HGA), which has been found as a co-infection with *B. divergens* in reservoir mammals [67,73,87]. *Anaplasma phagocytophilum* may induce immunosuppression in hosts [88], perhaps facilitating co-infections in the animal reservoirs not found in the tick due to a decrease in fitness of the tick vector when infected with both TBPs. Further investigations are necessary to unravel the underlying mechanism of this remarkable finding.

Although clade X *Babesia*, mainly *B. divergens*, has been implied as the cause of most European human babesiosis [12,89], no Dutch cases have been reported to date. Based on the findings of this study, this could be partially explained by the limited spread of the reservoir hosts *C. elaphus* and *B. bonasus*, meaning human exposure to infected ticks and consequently disease risk are low. Another possible explanation is that *B. divergens* strains from cattle and deer differ in their zoonotic potential [12]. It is relevant to note that just as tick populations have been expanding in recent decades [90,91], so has the deer population in Europe, with an increase of 54% in *C. elaphus* population between 1984 and 2000 [92]. Therefore, surveillance of *Babesia* in ticks and free ranging vertebrates remains useful for monitoring its future distribution.

*Babesia* clade I species *B. microti*, *B. vulpes* and three different *Babesia* badger-type haplotypes were found in this study. *Babesia vulpes* was detected in *N. procyonoides* and in the widely distributed *V. vulpes*. Although not zoonotic, this *Babesia* species has been known to cause disease in dogs [9,56,93,94,95] and has not been previously described in the Netherlands [9]. Likewise, *Babesia* badger-type haplotypes, which were found in this study in *M. meles* in accordance with previous publications [96,97], have been implied in symptomatic babesiosis in dogs [6].

Based on 18 S rRNA sequences *B. microti* was detected in bank voles (*M. glareolus*) (7%) and wood mice (*A. sylvaticus*) (0.7%), which supports their involvement in the natural life cycle of *B. microti* [98,99]. Its presence has been reported in *A. sylvaticus* in England with a prevalence of 8.8% [100]. However, *A. sylvaticus* was found to be negative for this parasite in most of the other studies from Europe [98,99,101]. The absence or low parasite load of *A. sylvaticus* may be due to the preference of this species for an open habitat leading to a lower tick load [101]. In contrast *B. microti* has been repeatedly detected across Europe in *M. glareolus* with a prevalence ranging from 0.03% to 40% in Germany and England respectively [99,102].

According to data obtained from the NDFF (the Dutch national database of flora and fauna), both bank voles and wood mice are widely distributed throughout the Netherlands. However, in our study the prevalence in a large cohort of questing ticks was relatively low (<3%). The rate of infected ticks in other European countries has a wide range, from 0.1% to 50.8% in Germany and Poland respectively [103,104]. This large variation in prevalence has been attributed to seasonal differences, patchy distribution of *Babesia* spp., the short lifespan of the hosts, the differences in the sensitivity of the detection methods and the selection of collection technique (from an animal host or the vegetation) [79,105,106].

The seroprevalence of *B. microti* in humans was reported to be 1% in eastern Croatia [107], 1.5% in Switzerland [32], 9% in Belgium [35], and 13% in Germany [108], inferring some level of human exposure in European countries [98]. Nonetheless, babesiosis cases caused by *B. microti* are rare in Europe, as opposed to North America where it is the causative agent of most human piroplasmosis cases [106], possibly due to the lower pathogenicity of the various European strains [12,32,101].

Another layer of complexity is added by the shared sylvatic cycle of *B. microti* with some *B. burgdorferi* sl genospecies and *N. mikurensis* [61]. Our study shows these TBPs co-occurred in questing nymphs significantly more than expected by chance, reflecting their ecological overlap. Co-infections between two or more of these TBPs have previously been reported both in questing ticks [109,110,111] and in ticks feeding on humans [39]. Such co-infections have been hypothesized to affect the course of disease [109,112,113]. Moreover, co-infections of other TBPs with *B. burgdorferi* sl might go undetected under the diagnosis of Lyme disease.

This and many other studies assess the presence of *Babesia* DNA, not its viability or infectivity. However, the inability of DNA-based detection methods to asseverate infectiousness does not render the approach irrelevant for surveillance of *Babesia*, especially when implemented in screening *I. ricinus* ticks, which have been widely implicated as the primary vectors for *Babesia* in Europe [24,42].

Routinely monitoring the potential wildlife hosts of zoonotic pathogens and their vectors together with insights into the genetic diversity and enzootic cycles of European strains may help effective and timely response to emerging zoonoses [58,114].

## 4. Materials and Methods

### 4.1. Collection of Field Samples

The collection of questing *I. ricinus* relied predominantly on convenience sampling from previous and ongoing studies in the Netherlands [61,62,115,116]. Spleen samples from wildlife were collected from culled or deceased animals and were either sent to the Dutch National Institute for Public Health and the Environment (RIVM) directly or via the Dutch Wildlife Health Centre (DWHC). The collection of spleen samples from mustelids and *Sciurus vulgaris* were described in previous studies [117,118]. Spleens and livers of hunted red foxes (*V. vulpes*) were opportunistically collected in a study for detection of fox tape worm [119]. Spleens and livers of hunted and road-killed raccoon dogs (*N. procyonoides*) were opportunistically collected in a study on zoonotic pathogens of raccoon dogs [120]. Moles (*Talpa europaea*) were culled by the employees of the Water Board as part of their control task, and were tested at the RIVM. Blood samples of wild boar were collected by hunters in 2014 and serum was sent to the RIVM as part of an ongoing surveillance of animal diseases, which was coordinated by GD Animal Health [121]. Spleens were collected from 99 birds that were found dead or were euthanized and sent to the DWHC for postmortem examination. These birds were identified as *Chloris chloris* (n = 7), *Coccothraustes coccothraustes* (n = 2), *Coloeus monedula* (n = 6), *Fringilla coelebs* (n = 3), *Garrulus glandarius* (n = 1), *Parus major* (n = 5), *Phylloscopus trochilus* (n = 1), *Pica pica* (n = 4), *Pyrrhula pyrrhula* (n = 1), *Sturnus vulgaris* (n = 7), *Turdus iliacus* (n = 5), *Turdus*
*merula* (n = 49), and *Turdus philomelos* (n = 8). The collection of EDTA-blood from *C. capreolus* was described in a previous study [122]. EDTA-blood from livestock (*B. bonasus, Bos taurus, Equus ferus caballus,* and *O. aries*) grazing in or adjacent to nature areas was collected by qualified veterinarians for medical purposes. Spleens from rodents were collected in different studies on the ecology of tick-borne pathogens in the Netherlands from 13 locations between August–October 2018 and March–June 2019. Rodent trapping, anesthetization, euthanasia, and all other aspects of the animal experiments were approved by the Central Committee Animal Experimentation in the Netherlands (AVD1040020173624), the Animal Welfare Body of Wageningen University (2017.W-0049), and the Netherlands Ministry of Economic Affairs (FF/75A/2015/014).

Tick species and stages were identified morphologically using stereo-microscope and identification keys [123,124]. All samples were kept frozen (−20 °C or −80 °C) until further processing. Most ticks and tissue samples were collected as convenience sampling. This means there is no congruous tempo-spatial context to the data. This allowed the study to comprise a vast amount of data for the evaluation of the research question. Moreover, ticks from Antwerpen collected as part of an urban tick study in Belgium [62] were included and deemed relevant for this study because of their geographic closeness.

### 4.2. DNA Extraction, qPCR, and RLB Protocols

The sampling approach meant that for questing nymphs two different molecular detection methods were employed: RLB and qPCR. RLB provides higher specificity, and the ability to detect mixed infections [125,126,127], but overtime was substituted by a more efficient and robust qPCR assay. Because of their differences, this study does not compare the infection prevalence between surveys carried out with different molecular techniques (See Table 2 and Table 3). DNA from questing ticks was extracted by alkaline lysis [128]. DNA from engorged ticks, blood and spleen samples was extracted using the DNeasy^®^ Blood & Tissue kit (QIAGEN, Hilden, Germany) as per the manufacturer’s instructions. To detect potential cross-contamination, negative controls were included in each batch of extraction. DNA lysates were screened for the presence of *Babesia* species from the *B.* sensu stricto clade with a qPCR targeting a 62-bp portion of the 18 S rRNA gene [129]. *Babesia* species from the *B. microti-like* clade were detected using a qPCR, which targets a 104-bp fragment of the internal transcribed spacer (ITS) region using primers 5’-CTCACACAACGATGAAGGACGCA-3’ (Bmicr_ITS_F), 5’-AACAGAGGCAGTGTGTACAATACATTCAGA-3’ (Bmicr_ITS_R), and the probe 5-Atto 520-GCA +GAATTTAG+CAAAT+CAACAGG-BHQ-1-3’ (Bmicr_ITS_px1). These qPCRs were carried out on a LightCycler 480 (Roche Diagnostics Nederland B.V, Almere, the Netherlands) in a final volume of 20 μL with iQ multiplex Powermix, 3 μL of sample and 0.2 μM for all primers and different concentrations for probes [130]. The presence of *Babesia* spp. in tick lysates from some studies was determined by PCR followed by RLB as described before [16,60]. Plasmid and negative water controls were used on every plate tested. To minimize contamination, and false-positive samples, the DNA extraction, PCR mix preparation, sample addition, and qPCR analyses were performed in separated air locked dedicated labs.

### 4.3. DNA Sequencing

Samples that were found positive by qPCR were amplified by conventional PCR, targeting a 400 to 440-bp fragment of the 18 S rRNA gene [16] and a conventional PCR targeting a fragment of the COI gene (Table 5). Moreover, 18 S rDNA is the most common marker used for classification at higher taxonomic levels due to its high sensitivity and a wide range of target species [2]; however, it has some limitations for discrimination at species (or lower) level of piroplasmids and for studies on intraspecific variability because of insufficient sequence variation among closely related species and the presence of several different gene copies [131,132].

The fragment of the mitochondrial COI gene shows high interspecific variability and discrimination accuracy, on the other hand it has some drawbacks such as having higher A + T content [131,133] and lower efficiency of amplification. Thus, both approaches were used in order to type the samples as accurately as possible.

Positive controls of DNA samples from *B. divergens* and *B. microti were* kindly supplied by Dr. Simone Caccio (Istituto Superiore di Sanità, Rome). The COI fragments of *Babesia* from the *B. microti*-clade were amplified as described previously [134], whereas the *Babesia* sensu stricto clade sequences were obtained by using 5’-ATWGGATTYTAT ATGAGTAT-3’ (Cox1_Bab_For1) and 5’-ATAATCWGGWATYCTCCTTGG-3’ (Cox1_Bab_Rev1) as primers [7]. PCR products were sequenced by dideoxy-dye termination sequencing of both strands, and compared with sequences in GenBank (http://www.ncbi.nlm.nih.gov/ accessed on 3 January 2019), using BLAST [135].

### 4.4. Species Identification Based on DNA Sequencing

The sequences were stored and analyzed in BioNumerics (Version 7.1, Applied Math, Sint-Martens-Latem, Belgium) after subtraction of the primer sequences for species identification. The DNA sequences that were generated in this study can be found in Appendix A.

The collected sequences were aligned with those from related organisms in GenBank, and pairwise alignments were generated using unweighted pair group method with arithmetic mean (UPGMA). A species was assigned to a sequence, which was identical or more than 99% similar to the reference sequences from GenBank. Extra attention was given to the 18 S rRNA fragment, where *B. capreoli* (FJ944828) is only two nucleotides different from *B. divergens* (AY046576). Sequences with double peaks in some of the sequence trace files (ambiguous nucleotides) at critical positions were not assigned to the species level.

### 4.5. Species Specific PCR

Infections with more than one *Babesia* from the *Babesia* sensu stricto clade were suspected in *C. capreolus* and *C. elaphus* samples on the basis of double peaks in some of the sequence trace files. To be able to quantify the occurrence of mixed infections and to identify the *Babesia* species involved, species-specific primers were designed for *B. venatorum*, *B. divergens*, *Babesia* sp. deer clade, and *B. capreoli*. These primers (100 nM each) specifically amplified small fragments of the COI gene (Table 5) using the following PCR program: 15 min at 95 °C, 35 cycles each consisting of 30 s at 95 °C, 30 s at 60 °C, and 1 min at 72 °C, and ending with 10 min at 72 °C. The HotStarTaq Polymerase Kit (Qiagen, Venlo, the Netherlands) was used for all PCR experiments. PCR products were detected by electrophoresis in a 1.5% agarose gel stained with SYBR gold (Invitrogen, Leiden, the Netherlands).

### 4.6. Data Collection on Geographic Spread of Vertebrates in The Netherlands

The geographic distribution of reservoir hosts for the identified *Babesia* species was retrieved from https://www.verspreidingsatlas.nl on 4 September 2020 [137], setting the time frame between the year 2000 and 2020. The geographic distribution of these vertebrates is based on field observations made by trained volunteers from the Dutch Mammal Society and are registered in 1 km^2^ blocks, out of 41,543 km² in the Netherlands.

### 4.7. Statistical Analyses

In order to test the relation between *Babesia* clade I or *Babesia* clade X with other TBPs detected in questing ticks, the significance of the observed (O) co-infections versus the randomly expected (E = (O−E)2/E) co-infections was assessed by the Fisher’s exact test and p-values (*p* < 0.05) were corrected using the Bonferroni procedure for multiple testing. The analyses were conducted in R [138] using the base package, tidyverse and dplyr [139,140].

## Figures and Tables

**Figure 1 pathogens-10-00386-f001:**
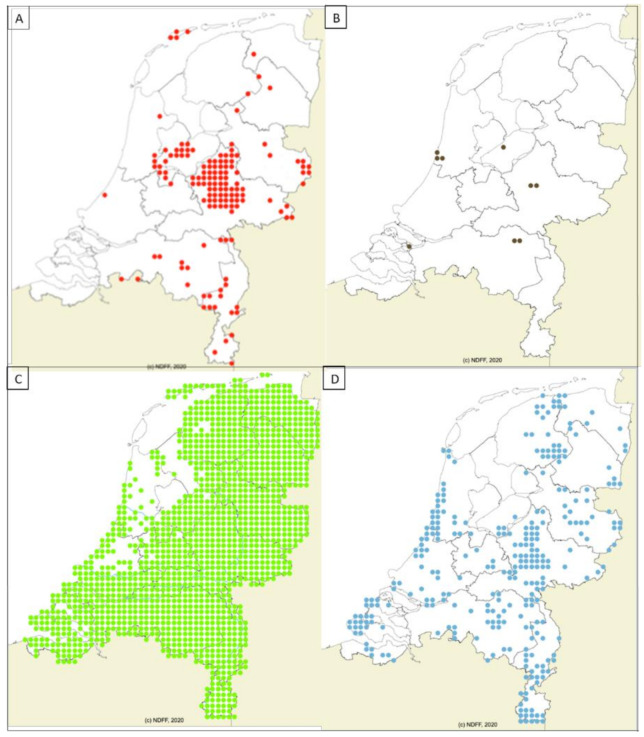
Distribution of wild competent wildlife species for *Babesia* clade X species found in the Netherlands. (**A**) *Cervus elaphus* (Red deer), (**B**) *Bison bonasus* (European bison), (**C**) *Capreolus capreolus* (Roe deer), (**D**) *Dama dama* (Fallow deer). Source: https://www.verspreidingsatlas.nl (accessed on 4 September 2020).

**Figure 2 pathogens-10-00386-f002:**
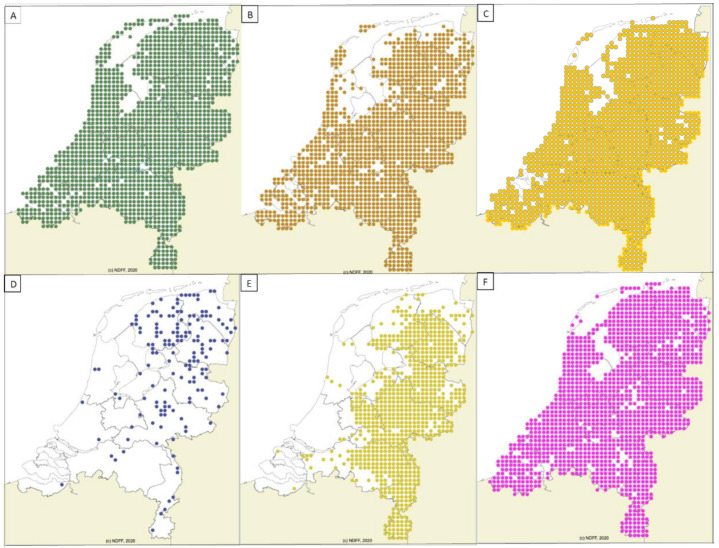
Distribution of competent wildlife species for *Babesia* clade I species found in the Netherlands. (**A**) *Apodemus sylvaticus* (Woodmouse), (**B**) *Myodes glareolus* (Bank vole), (**C**) *Vulpes vulpes* (Red fox), (**D**) *Nyctereutes procyonoides* (Raccoon dog), (**E**) *Meles meles* (European badger), (**F**) *Microtus arvalis* (Common vole). Source: https://www.verspreidingsatlas.nl (accessed on 4 September 2020).

**Table 1 pathogens-10-00386-t001:** Presence of *Babesia* species in vertebrate tissue samples.

Vertebrate (Order)	Vertebrate (Species)	Tested (n)	Clade I Positive (%)	Clade X Positive (%)	Samples Successfully Typed (18 S rRNA Typing)	>99% Identity (GenBank)
Artiodactyla	*Bos taurus*	116	0	0	-	
	*Bison bonasus*	19	0	4 (21%)	*B. divergens* (2)	AY046576
	*Capra aegagrus*	5	0	0	*-*	
	*Capreolus capreolus*	608	0	518 (85%)	*B. capreoli* (291) *B. venatorum* (3) Ambiguous (23) **	AY726009 FJ215873
	*Cervus elaphus*	147	0	94 (64%)	*B. divergens* (26) *Babesia* sp. deer clade (32)	AY046576 HQ638138
	*Dama dama*	100	0	9 (9%)	*B. capreoli* (4) *Babesia* sp. deer clade*-*EU (1)	AY726009 HQ638138
	*Ovis aries*	634	0	10 (2%)	*B. divergens* (2) *B. venatorum* (3)	AY046576 FJ215873
	*Sus scrofa*	111	0	0	-	
Aves	13 species	99	0	0	-	
Carnivora	*Martes foina*	134	0	0	-	
	*Martes martes*	128	0	0	-	
	*Meles meles*	128	108 (84%)	0	*B.* badger*-*type A (48) *B.* badger*-*type B (4) *B.* badger*-*type C (1)	KT223484 KT223485 MG799847
	*Mustela putorius*	242	0	0	-	
	*Nyctereutes procyonoides*	7	2 (29%)	0	*B. vulpes* (2)	AF188001
	*Vulpes vulpes*	173	124 (72%)	0	*B. vulpes* (58)	AF188001
Eulipotyphla	*Talpa europaea*	125	0	0	-	
Erinaceidae	*Erinaceus europaeus*	32	0	0	-	
Lagomorpha	*Lepus europaeus*	150	0	0	-	
	*Oryctolagus cuniculus*	88	0	0	-	
Perissodactyla	*Equus ferus caballus*	15	0	0	-	
Rodentia	*Apodemus flavicollis*	29	0	0	-	
	*Apodemus sylvaticus*	634	5 (0.8%)	0	*B. microti* (2)	KX161765
	*Castor fiber*	8	0	0	-	
	*Microtus arvalis*	100	3 (3%)	0	-	
	*Myodes glareolus*	405	25 (6%)	0	*B. microti* (12)	KX161765
	*Ondatra zibethicus*	210	0	0	-	
	*Rattus rattus*	49	0	0	-	
	*Sciurus vulgaris*	115	0	0	-	

DNA extracts from spleen or EDTA-blood of 4611 wildlife and free ranging domesticated animals were tested by two qPCRs (Clade I and Clade X). Typing was performed on qPCR-positive samples, and was based on sequencing a fragment of the 18 S rRNA (see methods). Typing of qPCR-positive samples by 18 S rRNA PCR and sequencing was not always successful. ** not typeable due to double peaks at discriminatory nucleotide positions in trace files.

**Table 2 pathogens-10-00386-t002:** Presence of *Babesia* Clade X in Questing *I. Ricinus.*

Study Acronym	*I. ricinus*	Method	Tested	Positive		Samples Successfully Typed/Sequenced				Period	Province
(Reference)	stage		n	n	%	*B. capreoli*	*B. divergens*	*Babesia* sp. deer clade	*B. venatorum*	years	sites (n)
National Survey [44]	N + A	RLB	857	13	0.015				13	2000–2010	Gelderland (7)
	N + A	RLB	232	7	0.03				7	2006–2010	Limburg (1)
	N + A	RLB	1995	11	0.006		1		10	2000–2010	Noord-Holland (4)
	N + A	RLB	242	1	0.004				1	2006–2010	Brabant (1)
	N + A	RLB	393	4	0.01				4	2006–2010	Drenthe (2)
	N + A	RLB	232	0	0					2006–2010	Overijssel (2)
	N + A	RLB	162	2	0.012				2	2006–2010	Zuid-Holland (1)
	N	RLB	79	3	0.038				3	2007–2010	Friesland (1)
	N	RLB	6	0	0					2007	Groningen (1)
	N	RLB	40	2	0.05				2	2007–2010	Utrecht (1)
Duin-Kruidberg [16]	L + N + A	RLB	1.488	14	0.009		1		13	2003–2007	Noord-Holland (1)
Lizard study [60]	N + A	RLB	491	2	0.004				2	2007–2009	Gelderland (8)
Drenthe *	N + A	RLB	1727	3	0.002				3	2010–2012	Drenthe (32)
Rodent study [61] *	N	qPCR	7637	73	0.01	5	2	1	55	2012–2014	Gelderland (2)
Urban ticks [62]	N	qPCR	1780	19	0.011	2			16	2014–2016	Antwerpen, Belgium (13)
	A	qPCR	268	3	0.011				3	2014–2016	Antwerpen, Belgium (11)
Cattle study [63]	N	qPCR	112	0	0					2015	Drenthe (1)
	N	qPCR	181	2	0.011				1	2015	Gelderland (1)
	N	qPCR	191	1	0.005				1	2015	Noord-Brabant (1)
	N	qPCR	735	1	0.001					2015	Noord-Holland (4)
	N	qPCR	165	2	0.012				1	2015	Overijssel (1)
	N	qPCR	783	12	0.015				8	2015	Utrecht (4)
	N	qPCR	185	1	0.005				1	2015	Zuid-Holland (1)
De Groote Peel *	N	qPCR	79	4	0.051				3	2017	Brabant (1)
Sheep study *	N + A	qPCR	1555	24	0.015	1			16	2017	Drenthe (3)
Microbiome *	N	qPCR	560	13	0.023				11	2018	Gelderland (1)
	N	qPCR	520	2	0.004				1	2018	Noord-Holland (1)
	N	qPCR	1960	47	0.024	3			33	2018	Utrecht (1)
Nijverdal *	N	qPCR	1194	223	0.187					2019	Overijssel (1)
Total			25849	489	1.9%	11	4	1	210	2000–2019	

Ticks from eleven studies were tested for the presence of *Babesia* clade I with either RLB or qPCR. Typing was done by RLB directly (n = 62) or conventional PCR and sequencing on a fragment of the 18 S rRNA gene on part of the qPCR-positive samples (n = 164). Stage (N/A) refers to the tick life stage, i.e., adult (A) or nymph (N). * Molecular detection *of Babesia* clade X was performed for this study. (n) Data of field sites from one study were combined to one data point per province.

**Table 3 pathogens-10-00386-t003:** Presence of *Babesia* clade I in questing *I. ricinus.*

Study Acronym	*I. ricinus*	Method	Tested	Positive		Typing	Period	Province
(Reference)	stage		n	n	%	Sequencing	years	sites (n)
National Survey [44]	N + A	RLB	857	6	0.7	-	2000–2010	Gelderland (7)
	N + A	RLB	1995	2	0.1	-	2000–2010	Noord-Holland (4)
	N + A	RLB	242	0	0	-	2006–2010	Brabant (1)
	N + A	RLB	393	0	0	-	2006–2010	Drenthe (2)
	N + A	RLB	232	0	0	-	2006–2010	Limburg (1)
	N + A	RLB	232	8	3.4	-	2006–2010	Overijssel (2)
	N + A	RLB	162	0	0	-	2006–2010	Zuid-Holland (1)
	N	RLB	79	1	1.3	-	2007–2010	Friesland (1)
	N	RLB	6	0	0	-	2007	Groningen (1)
	N	RLB	40	0	0	-	2007–2010	Utrecht (1)
Duin-Kruidberg [16]	N + A	RLB	908	2	0.20	*B. microti* (2)	2003–2007	Noord-Holland (1)
Lizard study [60]	N + A	RLB	491	6	1.20	-	2007–2009	Gelderland (8)
Drenthe (this study)	N + A	RLB	1727	0	0	-	2010–2012	Drenthe (32)
Rodent study [61]	N	qPCR	7637	393	5.1	*B. microti* (39)	2012–2014	Gelderland (2)
Cattle study [63] *	N	qPCR	112	0	0	-	2015	Drenthe (1)
	N	qPCR	181	14	7.7	*B. microti* (4)	2015	Gelderland (1)
	N	qPCR	191	0	0	-	2015	Noord-Brabant (1)
	N	qPCR	735	0	0	-	2015	Noord-Holland (4)
	N	qPCR	165	0	0	-	2015	Overijssel (1)
	N	qPCR	783	0	0	-	2015	Utrecht (4)
	N	qPCR	185	0	0	-	2015	Zuid-Holland (1)
De Groote Peel *	N	qPCR	79	0	0	-	2017	Brabant (1)
Nijverdal *	N	qPCR	1194	50	4.20	-	2019	Overijssel (1)
Total			18626	482	2.6	*B. microti* (45)	2000–2019	

Ticks from eight studies were tested for the presence of *Babesia* clade I with either RLB or qPCR. Typing was done by conventional PCR and sequencing on a fragment of the 18 S rRNA gene on part of the positive samples (n = 60) and 45 isolates yielded a DNA sequence identified as *B. microti.* Stage (N/A) refers to the tick life stage, i.e., adult (A) or nymph (N). * Molecular detection *of Babesia* clade I was performed for this study. (n)Data of field sites from one study were combined to one data point per province.

**Table 4 pathogens-10-00386-t004:** Coinfection of tick-borne pathogens with *Babesia species.*

*Babesia* Clade	Pathogen	Positive	Observed	Expected	*P*-Value	Odds Ratio
*Babesia* Clade I	*B. burgdorferi* sl	1292	273	65	1.10 × 10^−15^	11.72
	*B. miyamotoi*	261	14	13	1	1.08
	*N. mikurensis*	939	129	47	1.10 × 10^−15^	3.86
	*A. phagocytophilum*	642	21	32	1.80 × 10^−1^	0.62
	*Babesia* Clade X	123	7	6	1	1.15
*Babesia* Clade X	*B. burgdorferi* sl	1292	17	18	1	0.93
	*B. miyamotoi*	261	6	4	0.90	1.69
	*N. mikurensis*	939	6	13	0.18	0.42
	*A. phagocytophilum*	642	2	9	0.04	0.2
	*Babesia* Clade I	442	7	6	1	1.15

A total of 8831 *I. ricinus* nymphs from Nijverdal and the rodent study (Table 3 and Table 4) were tested for other tick-borne pathogens. A Fisher’s exact test was used in order to evaluate the significance of the observed number of co-infections between either Babesia clade I or Babesia clade X and each tick-borne pathogen, and what would be randomly expected. This was assessed by calculating the Odds Ratio and their 95% confidence intervals (not shown). *p*-values were corrected using the Bonferroni test.

**Table 5 pathogens-10-00386-t005:** Nucleotide sequences of primers and probes used in this study.

*Babesia*	Target	Primer (Name)	Primer (Sequence)	Purpose	Size (bp)	Reference
Clade I	ITS	Bmicr_ITS_F	5’-CTCACACAACGATGAAGGACGCA-3’	qPCR	103 bp	[136]
		Bmicr_ITS_R	5’-AACAGAGGCAGTGTGTACAATACATTCAGA-3’			
		Bmicr_ITS_Px1	5′- Atto520-GCA+GAATTTAG+CAAAT+CAACAGG- BHQ1-3′			
Clade X	18SrRNA	Bab_18SrRNA-F	5’-CAGCTTGACGGTAGGGTATTGG-3’	qPCR	62 bp	[129]
		Bab_18SrRNA-R	5’-TCGAACCCTAATTCCCCGTTA-3’			
		Bab_18SrRNA-P	5’-Atto647N-CGAGGCAGCAACGG-MGB-BHQ2-3’			
*Babesia* spp.	18SrRNA	Bath-Fn	5’-TAAGAATTTCACCTCTGACAGTTA-3’	PCR/SEQ	±420 bp	[16]
		Bath-Rn	5’-ACACAGGGAGGTAGTGACAAG-3’			
Clade I	COI	Cox1F133	GGAGAGCTAGGTAGTAGTGGAGATAGG	PCR/SEQ	1023 bp	[134]
		Cox1R1130	GTGGAAGTGAGCTACCACATACGCTG			
Clade X	COI	Cox1_Bab_For1	5′-ATWGGATTYTATATGAGTAT-3′	PCR/SEQ	±1250 bp	[7]
		Cox1_Bab_Rev1	5′-ATAATCWGGWATYCTCCTTGG-3′			
*B. venatorum*	COI	Bven-F159	5’-ATTGGAAGTGGTACTGGTTGGACTT-3’	PCR	538 bp	This study
		Bven-R696	5’-GACATCATTACGATTCCTATGC-3’			
*B. divergens*	COI	Bdiv-F165	5’-AGTGGAACTGGGTGGACATTGTAC-3’	PCR	234 bp	This study
		Bdiv-R398	5’-TACCGGCAATGACAAAAGTAG-3’			
*B. capreoli*	COI	Bcap-F165	5’-AGTGGAACAGGATGGACGCTATAT-3’	PCR	443 bp	This study
		Bcap-R607	5’-GTCTGATTACCGAACACTTCC-3’			
*Babesia* sp. deer clade	COI	Bodo-F360	5’-CTTTGACTGCTTTCTTGTTG-3’	PCR	434 bp	This study
		Bodo-R793	5’-ATCATAACAATTCCTATGCTC-3’			

The two qPCRs were used for the screening of wildlife and tick samples for the presence of *Babesia spp.* qPCR-positive samples were further analyzed by conventional PCRs and sequencing for species identification (PCR/SEQ). For the detection of multiple *Babesia* sensu stricto species (Clade X) in one sample (mixed infections), four conventional PCRs, each specifically targeting one *Babesia* species, were performed and analyzed by TBE-agarose gel electrophoresis. Abbreviations: Atto520 fluorescent dye; BHQ1 (blackholequencher) quenches the fluorescence of Atto520; “+” stands for Locked Nucleic Acid (LNA), this is used to raise the annealing-temperature of the probe; MGB stands for Minor groove binder, which is also used for raising the annealing-temperature.

## Data Availability

The data presented in this study are available within the manuscript and Appendix A.

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
