# Peer review of "Circulation of Babesia Species and Their Exposure to Humans through Ixodes ricinus"

_pathogens, 2021, doi:10.3390/pathogens10040386_

Round 1
Reviewer 1 Report
Dear authors,
Circulation of Babesia species and their exposure to humans through Ixodes ricinus is a well written and interesting article.
There remain some issues at this time and authors should consider the comments useful for further revision of the manuscript. Importantly, there are many grammatical errors throughout the manuscript and so it has to read by a native speaker for clarity. It is very important to fix all the grammatical errors before resubmitting the manuscript.
Comments :
What do authors mean by COI gene?
What is the basis of selecting 18S rDNA as a Reference gene? For Babesia microti, it is important to include Cox gene as a reference gene
Line 37: What do authors mean by autochthonous Babesisois ? Please explain
Table 2 & 3, authors write N and A, it is important to write expanded forms in the figure legend.
It is important to make an abbreviation table as there are lot of short terms throughout the manuscript
In the discussion section, authors write NDEF. Please expand NDEF
It would be worth to split the tables to more tables to avoid confusion
Table 4: Coinfection of other tick borne pathogens- what do authors mean by observed and expected? It is important to specify about these terms
Reviewer 2 Report
The aim of the current manuscript was to evaluate reservoir hosts and Ixodes ricinus as a vector of Babesia spp. of zoonotic potential in Netherland. A cross-sectional study was carried out. In total, 4,611 tissue samples from 41 mammalian species and 13 bird species, and 25,849 questing ticks of I. ricinus were included in the comprehensive study that yielded abundant data. The manuscript is well written with the conclusions drew on the solidly presented data. It is suitable for this journal. This reviewer has only a few comments on minor issues.
First, scientific names of genus and species should be fully spelled out first time they appear in text including abstract, and afterwards the name of genus is shortened with a capital initial followed by a period “.” These names should always be italicized. There are several names not following these rules throughout the manuscript. Please make necessary changes.
A review article on human babesiosis has just been published: https://www.mdpi.com/2076-2607/9/2/440, which should be helpful in your introduction of human babesiosis.
A period “.” rather than a comma “,” should be used in numbers with decimals. There are at least a couples of cases a comma is used. Please check the entire manuscript for the misuse.
Line number: No numbers are used for M&M section up to 4.4, and 4.4 starts from number 1, which causes confusion.
Section 4.2: Some primer sequences have “+”. What does it mean? In addition, spaces are included, which should be removed. PCR and RLP can be used once they have defined. Check the entire manuscript for abbreviation use. Make sure they are defined when they first appear.
L69: Piroplasmida is not a scientific species name and should not be italicized.
L103: Ixodidae is a family name, and should not be italicized.
L108: instar is a term used for insects, especially immature stages of flies. Is it probable using in arachnids like ticks? I would not use it in the way as it is used here.
L119: reformat the references.
